# Modular Clinical Decision Support Networks (MoDN)—Updatable, interpretable, and portable predictions for evolving clinical environments

**Cécile Trottet**[1], **Thijs Vogels**[1], **Kristina Keitel**[2], **Alexandra V. Kulinkina**[3,4], **Rainer Tan**[5,6,7], **Ludovico Cobuccio**[5], **Martin Jaggi**[8], **Mary-Anne Hartley**[1,9]*

**1** Intelligent Global Health Research Group, Machine Learning and Optimization Laboratory, Swiss Federal Institute of Technology (EPFL), Lausanne, Switzerland, **2** Division of Pediatric Emergency Medicine, Department of Pediatrics, Inselspital, Bern University Hospital, University of Bern, Switzerland, **3** Digital Health Unit, Swiss Center for International Health, Swiss Tropical and Public Health Institute, Allschwil, Switzerland, **4** University of Basel, Basel, Switzerland, **5** Clinical Research Unit, Swiss Tropical and Public Health Institute, Allschwil, Switzerland, **6** Ifakara Health Institute, Ifakara, Tanzania, **7** Center for Primary Care and Public Health (Unisanté), Lausanne, Switzerland, **8** Machine Learning and Optimization Laboratory, Swiss Federal Institute of Technology (EPFL), Lausanne, Switzerland, **9** Laboratory of Intelligent Global Health Technologies, Biomedical Informatics and Data Science, Yale School of Medicine, New Haven, CT, USA

☯ These authors contributed equally to this work.

* mary-anne.hartley@epfl.ch

**Data Availability Statement:** Anonymized data are publicaly available here: https://zenodo.org/record/400380#.Yug5kuzP00Q The full code is available at

## Abstract

Clinical Decision Support Systems (CDSS) have the potential to improve and standardise care with probabilistic guidance. However, many CDSS deploy static, generic rule-based logic, resulting in inequitably distributed accuracy and inconsistent performance in evolving clinical environments. Data-driven models could resolve this issue by updating predictions according to the data collected. However, the size of data required necessitates collaborative learning from analogous CDSS's, which are often imperfectly interoperable (IIO) or unshareable. We propose Modular Clinical Decision Support Networks (MoDN) which allow flexible, privacy-preserving learning across IIO datasets, as well as being robust to the systematic missingness common to CDSS-derived data, while providing interpretable, continuous predictive feedback to the clinician. MoDN is a novel decision tree composed of feature-specific neural network modules that can be combined in any number or combination to make any number or combination of diagnostic predictions, updatable at each step of a consultation. The model is validated on a real-world CDSS-derived dataset, comprising 3,192 paediatric outpatients in Tanzania. MoDN significantly outperforms 'monolithic' baseline models (which take all features at once at the end of a consultation) with a mean macro $F_1$ score across all diagnoses of 0.749 vs 0.651 for logistic regression and 0.620 for multilayer perceptron ($p < 0.001$). To test collaborative learning between IIO datasets, we create subsets with various percentages of feature overlap and port a MoDN model trained on one subset to another. Even with only 60% common features, fine-tuning a MoDN model on the new dataset or just making a composite model with MoDN modules matched the ideal scenario of sharing data in a perfectly interoperable setting. MoDN integrates into consultation logic

the following GitHub repository: https://github.com/epfl-iglobalhealth/PLOSDH-MoDN-TrottetVogels2022.

**Funding:** This work took place within the framework of the DYNAMIC project that is funded by the Fondation Botnar, Switzerland (grant n° 6278), MAH received a subgrant for this work. The funders had no role in study, analysis, decision to publish, or preparation of the manuscript.

**Competing interests:** The authors have declared that no competing interests exist.

by providing interpretable continuous feedback on the predictive potential of each question in a CDSS questionnaire. The modular design allows it to compartmentalise training updates to specific features and collaboratively learn between IIO datasets without sharing any data.

## Author summary

Clinical Decision Support Systems (CDSS) are emerging as a standard-of-care, offering probabilistic guidance at the bedside. Many deploy static, generic rule-based logic, resulting in inconsistent performance in evolving environments. Machine learning (ML) models could resolve this by updating predictions according to the collected data. However, traditional methods are often criticised as uninterpretable "black-boxes" and are also inflexible to fluctuations in resources: requiring retraining (and costly re-validation) each time a question is altered or added. We propose MoDN: a novel, interpretable-by-design, modular decision tree network comprising a flexible composition of question-specific neural network modules, which can be assembled in real-time to build tailored decision networks at the point-of-care, as well as enabling collaborative model learning between CDSS with differing questionnaire structures without sharing any data.

## Introduction

Probabilistic decision-making in medicine has the potential to bypass costly and invasive clinical investigations and holds particular promise to reduce resource consumption in low-income settings [1]. However, it is impossible for any clinician to memorise the increasingly complex, evolving, and sometimes conflicting probabilistic clinical guidelines [2], which has driven the need for Clinical Decision Support Systems (CDSS) that summarise guidance into simple rule-based decision trees [3–5]. The digitalization of some commonly used CDSS into mobile apps has shown promise in increasing access and adherence to guidelines while laying the foundation for more systematic data collection [6–10].

Despite their promise to bridge the 'know-do gap', a surprisingly low proportion of popular guideline recommendations are backed by high-quality evidence ('know') [11], and an even lower proportion of mobile tools have been rigorously tested in practice ('do') [12–14]. While likely to generally improve and standardise care, their static and generic logic would result in inconsistent performance in light of changing epidemiology as well as an inequitable distribution of accuracy in underrepresented populations [1].

To address these limitations, there is a move toward data-driven predictions that incorporates machine learning (ML), [15–18] with the goal of leveraging more complex multi-modal data and derive self-evolving algorithms [19]. The WHO SMART guidelines [20] advocate for a faster and more systematic application of digital tools. More specifically, the last layer of these guidelines reflects the use of dynamic, big-data-driven algorithms for optimised outcomes and updatable recommendations. This move is predicted to improve CDSS safety and quality [21], especially in low-resource settings [22, 23]. Additionally, as such models improve with the addition of good quality data, they inherently incentivise better data collection, a positive feedback termed the *CDSS Loop* [24].

Regardless, the data collected with decision tree logic is fundamentally flawed by biased missingness [25]. Patients are funneled into high-yield question branches, creating systematic

missing values for questions that are not asked. Models trained on such data can easily detect patterns in the missingness of features rather than in their values, thus, not only failing to improve on the rule-based system, but also becoming clinically irrelevant. Further, when data is missing for different reasons than what was present in the training set, the model could be dramatically impacted.

Such issues of data quality and utility are secondary to the more general limitation of availability. Patient-level data is rarely shared due to well-considered concerns of privacy and ownership. The inability to share data fragments statistical power, compromises model fairness, [26] and results in poor interoperability, where CDSS users and developers do not align data collection procedures.

The latter can limit collaborative learning across analogous CDSS tools, restricting them to use only features that are available to all participants [15].

Even if the above issues of data quality, availability and interoperabilty are resolved (for example, with the powerful cross-EHR solution proposed by Google Research [19]), the 'updating' of ML models still poses a major regulatory issue which may make them unimplementable [27]. 'Perpetual updating' is one of the main motivations for the transition to ML [18] to combat 'relevance decay' [28], which can be significant in rapidly evolving environments with changing epidemiology and unreliable resources. Indeed, after the laborious processes of data collection, cleaning, harmonisation, model development and clinical validation, the tool could already be outdated. The problem is that while ML models can learn autonomously from updating data streams, each update requires whole-model retraining that invalidates the previous version, which is likely to make the promise of perpetual updates unfeasible.

In this work, we propose the Modular Decision Support Network (MoDN) to provide dynamic probablistic guidance in decision-tree based consultations. The model is flexibly extended during the course of the consultation, adding any number or combination of neural network *modules* specific to each question asked. This results in a dynamic representation of the patient, able to predict the probability of various diagnoses at each step of a consultation.

We validate MoDN on a real world CDSS-derived data set of 3000 pediatric outpatient consultations and show how the feature-wise modular design addresses the issues above, such as collaborative learning from imperfectly interoperable (IIO) datasets with systematic missingness, while improving data availability, model fairness and interpretability. The feature-wise modularisation makes MoDN interpretable-by-design, whereby it aligns its learning process with the clinician, in step-by-step "consultation logic". It can thus provide continuous feedback at each step, allowing the clinician to directly assess the contribution of each question to the prediction at the time of asking it. A major contribution, is the possibility to compartmentalise updates to affected features, thus retaining validity.

## Materials and methods

Below, we describe MoDN, a deep learning CDSS composed of interchangeable, feature-specific neural network modules. We validate it on a real-world CDSS-derived data set and visualise its capacity to represent patient groups and predict multiple diagnoses at each stage of the consultation. Two additional experiments are designed to test its portability to imperfectly interoperable data sets, and compartmentalise updates to a targeted feature sub-set.

### Data set

**Cohort description.** We train, test and validate MoDN on a CDSS-derived data set comprising 3,192 pediatric (aged 2–59 months) outpatient consultations presenting with acute febrile illness. The data was collected in nine outpatient departments across Dar es Salaam,

Tanzania between 2014 and 2016 as part of a randomised control trial assessing the effect of digital CDSS on antibiotic use [29]. The data had over 200 unique feature sets of asked questions (i.e. unique combinations of decision branches in the questionnaire).

**Ethics.** Written informed consent was obtained from the caregivers of all participants as described in Keitel et al. [29]. The study protocol and related documents were approved by the institutional review boards of the Ifakara Health Institute and the National Institute for Medical Research in Tanzania, by the Ethikkommission Beider Basel in Switzerland, and the Boston Children's Hospital ethical review board. An independent data and safety monitoring board oversaw the study. The trial was registered in ClinicalTrials.gov, identifier NCT02225769.

**Features and targets.** A subset of eight diagnoses and 33 features were selected in order to ensure interpretable reporting and limit computational cost. Selection of both targets and features were based primarily on prevalence (i.e. retaining the most prevalent). Features were additionally tested for predictive redundancy in bivariate Pearson's correlation, and strongly collinear features were randomly dropped. The features comprise demographics, medical history, clinical signs and symptoms and laboratory results collected at the time of consultation and are detailed in S1 Table.

MoDN aims to simultaneously predict eight retrospectively derived diagnoses, namely anaemia, dehydration, diarrhoea, fever without source, malaria, malnutrition, pneumonia, and upper respiratory tract infection. Patients could have none or several of these diagnoses.

It is also possible to predict (impute) any of the 33 missing features; experimentation on feature decoding is detailed in the supplement S1 Text.

**Pre-processing.** As explained, the decision tree logic of questionnaires in CDSS-derived data creates systematic missingness, where diagnostic endpoints may have unique feature sets. We exploit these patterns to derive the consultation logic (i.e. order of asked questions) and thus align training with clinical protocols. For groups of questions that are either all present or all missing according to the outcome, relative ordering is impossible, and thus randomised.

To ensure that our model performs well for patients outside of the training data set, we randomly partition the data into train ($n = 1914$, 60%), validation ($n = 639$, 20%), and test ($n = 639$, 20%) splits. We optimize the model on the training set only, tune its hyperparameters based on the validation split, and report its final performance on the test split that was not used in creating the model. We then obtain a distribution of estimates using five iterations of two-fold cross validation [30], where data is randomly re-partitioned.

## MoDN

**Model architecture.** MoDN comprises three core elements: *encoders*, *decoders*, and the *state* as listed below and summarised in Fig 1.

- The *state*, **s**, is the vector-representation of a patient. It evolves as more answers are recorded.

- *Encoders* are feature-specific and update the *state* with the value of a newly collected feature based on the current version of the *state*.

- *(Diagnosis) decoders* are output-specific and extract predictions from the *state* at any stage of the consultation. Predicted outputs can be any data set element, including the features themselves. These *feature decoders* provide a dynamic patient-specific imputation of values not yet recorded. All feature-decoding experimentation is detailed in the supplements S1 Text and in S1 and S2 Figs.

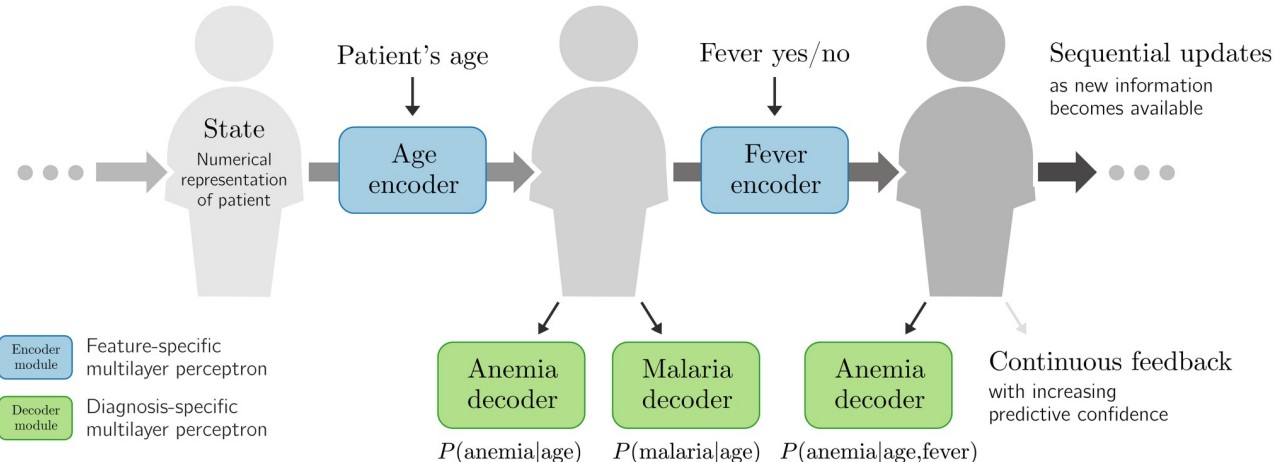

**Fig 1. The Modular Clinical Decision Support Network (MoDN).** The *state* is a representation of the patient, which is sequentially modified by a series of inputs. Here, we show in blue *age* and *fever* values as modifying inputs. Each input has a dedicated encoder which updates the *state*. At any point in this process, the clinician can either apply new encoders (to update the *state*) or decode information from the *state* (to make predictions, in green).

Encoders and decoders are thus respectively feature or output specific multilayer perceptrons (MLP). This modularises both the input space as well as the predictions made in the output space. More details are provided in the supplement S1 Text.

We consider the consultation data of a patient as an ordered list of (question, answer) pairs, $(q_1, a_1), (q_2, a_2), \ldots, (q_T, a_T)$. The ordering of questions asked simultaneously is randomized. As new information is being collected, the *state* vector $\mathbf{s} \in \mathbb{R}^s$ evolves as:

$$\mathbf{s}_0 = \mathcal{S}_0 \in \mathbb{R}^s, \text{ a trained constant}, \tag{1}$$

$$\mathbf{s}_t = \mathcal{E}_{q_t}(\mathbf{s}_{t-1}, a_t), \quad \text{for } t = 1 \ldots T, \tag{2}$$

where $\mathcal{E}_{q_t} : (\mathbb{R}, \mathbb{R}^s) \to \mathbb{R}^s$ is an *encoder* specific to question $q_t$. It is a small MLP with trainable parameters.

After gathering the first $t$ answers, the probability of a diagnosis $d$ is then predicted as:

$$p_t(d) = \mathcal{D}_d(\mathbf{s}_t), \tag{3}$$

where each *decoder* $\mathcal{D}_d : \mathbb{R}^s \to [0, 1]$ is also a small MLP with trainable parameters.

The algorithm is thus agnostic to the 1) input—accepting any type, number, or combination of inputs, 2) model architecture—where each encoder and decoder can be of any architecture, such as a CNN, MLP, etc., and 3) task—where any number of combination of task-specific decoders can be deployed at any point.

**Model optimisation.** We optimize the parameters of the trainable components of MoDN, written as calligraphic symbols, using a training set of $C$ completed consultations. In addition to a list of questions and answers, each consultation also features binary 'ground-truth' labels $y_d^c$ that indicate whether consultation $c$ was diagnosed with $d$. Following the principle of empirical risk minimisation, our training objective is a sum over $C$ consultations $c$, but also over $D$

potential diagnoses $d$, and $T$ 'time-steps':

$$\min \sum_{c=1}^{C} \sum_{d=1}^{D} \sum_{t=0}^{T} \ell(p_t^c(d), y_d^c) + R, \tag{4}$$

The parameters to be optimized are implicit in $p_t^c(d)$. For binary diagnoses, $\ell$ is cross-entropy loss. The inclusion of different time-steps in the objective ensures that MoDN can make predictions at any stage of the consultation.

The regularization term $R$ in our objective ensures that the *states* do not change more than necessary to encode the new information:

$$R = \frac{1}{s} \sum_{c=1}^{C} \sum_{t=1}^{T} \| \mathbf{s}_t^c - \mathbf{s}_{t-1}^c \|^2. \tag{5}$$

We optimize Eq 4 with the Adam optimizer [31]. For each step, we sample a batch of consultations, and sum the decoder losses at the multiple intermediate time-steps $T$. We randomize the order within blocks of simultaneously asked questions.

**Visualizing the MoDN *state*.** The *state* **S** may become highly dimensional in complex data sets rendering it uninterpretable. To gain some visual insight into this vectorised representation of a patient, we use the t-distributed stochastic neighbor embedding (t-SNE) [32] dimensionality reduction algorithm, where similar data points are mapped close to each other in a lower dimensional embedding. By overlaying the data points (*states*) with a colour representing the diagnosis, we can visualise how well the *state* represents the predicted label.

**Baseline models for MoDN performance.** MLP and logistic regression are used as baselines to compare with MoDN for binary diagnostic classification tasks (i.e. one model per diagnosis). Train-test splits and pre-processing is identical to MoDN with the exception of imputation. As traditional ML models cannot handle missing values, mean value imputation was performed. Performance is reported as macro F1 scores (the harmonic mean of precision and sensitivity). Models are compared with a paired $t$-test on a distribution of performance estimates derived from a $5 \times 2$ cross-validation as per Dietterich et al. [30]. We also report a calibration curve.

## Experimental set up

A common issue when CDSS are updated in light of newly available resources (e.g. new questions/tests added to the CDSS) is incomplete feature overlap between old and new data sets. To test the capacity of MoDN in such imperfectly interoperable (IIO) settings, we simulate IIO subsets within the 3,192-patient CDSS-derived data described previously. These IIO subsets are depicted in Table 1, where data sets $A$ and $B$ comprise 2,068 and 516 patients respectively. Performance is then evaluated on an independent test set of size 320 ($D$), in which all of the features are available. Internal validation for each model is performed on the remaining 288 patients ($C'$ and $C$ for validation of data sets $A$ and $B$ respectively, which differ in the number of features provided). Three levels of IIO (90%, 80% and 60%) are simulated between data sets $A$ and $B$ by artificially deleting random features in $A$. These are compared to a baseline of perfectly interoperable feature sets (100% overlap). Within these data sets, all experiments are performed with 5–fold randomisation of data set splits and available features to obtain a distribution of F1 scores which are averaged to a macro F1 score with 95% confidence interval.

**New IIO user experiment: Modularised fine-tune.** In this common scenario, a clinical site starts using a CDSS. It has slightly different resources and is thus IIO compared to more established implementation sites. However, the new site would still benefit learning from its

**Table 1. Imperfectly interoperable (IIO) data sets.** From the 3,192-patient CDSS-derived data set, we create two training sets with three levels of imperfect feature overlap (60, 80 and 90%) compared with perfect interoperability (100%). In our experiments, the owner of a small 'target' data set (fewer patients) wants to benefit from a larger 'source' data set without having access to this data. The 'source' may lack several features that are available in the 'target', yielding several levels of 'imperfect interoperability'. We construct validation sets with and without these missing features, as well as a held-out test set. The F1 scores we report in this paper are averages over five randomized folds of this data-splitting procedure.

| Split | Partition | Patients | Interoperable | Imperfectly interoperable |
|---|---|---|---|---|
| Train | Source (A) | 2 068 | Features | 60% features<br>80%<br>90% |
| | Target (B) | 516 | | |
| Validation | Source | 288 | | |
| | Target | 288 | | |
| Test | Source | 288 | | |

more established counterparts, while ensuring that the unique trends in its smaller, local data set are preserved. Due to ethical constraints, data sets cannot be shared.

We hypothesise that MoDN is able to handle this scenario via 'modularised fine-tuning' as depicted in Fig 2. Here, a MoDN is pre-trained on the larger (unseen) data set *A* and ported to *B* where all of the modules are fine-tuned. Thus personalising the existing modules as well as adding new modules unique to *B*, thus creating a new collaborative IIO model without sharing data.

**New IIO resource scenario: Modularised update.** In this common scenario, a site using a CDSS acquires new resources (e.g. new point-of-care diagnostic tools). It would like to update its CDSS model with the data collected from this new feature, but it cannot break the validity of the existing predictions that have been approved by the regulatory authority after a costly validation trial.

We hypothesise that MoDN is able to handle this scenario via feature-wise compartmentalisation of model updates. Keeping all experimental conditions identical as depicted in Fig 2, MoDN is pre-trained on the larger (unseen) data set *A* and ported to *B* (hosting new resources). The key difference is that no fine-tuning occurs. Rather, we seek to preserve the predictive validity of the MoDN modules of overlapping features by freezing them. Thus, the modules are first trained on A and then fixed, mimicking a validated model. When including data B, the modules are combined and only the encoders corresponding to the 'new' features in B are trained.

**Baseline models for IIO experiments.** Three baselines are proposed as depicted in blue, green and purple in Fig 2.

- **The static model** is where modules trained in *A* are directly tested in *B*, thus not considering additional IIO features.

- **The local model** is where modules are only trained on the target data set *B*, thus without insights from the larger source data set.

- **The global model** is the ideal, but unlikely, scenario of when all data can be shared between *A* and *B* and the modules are trained on the union of data ($A \cup B$).

## Results

For simplicity, only results for diagnostic decoders are reported. Results for a model including feature decoding and idempotence (i.e. where a specific question-answer pair will not change

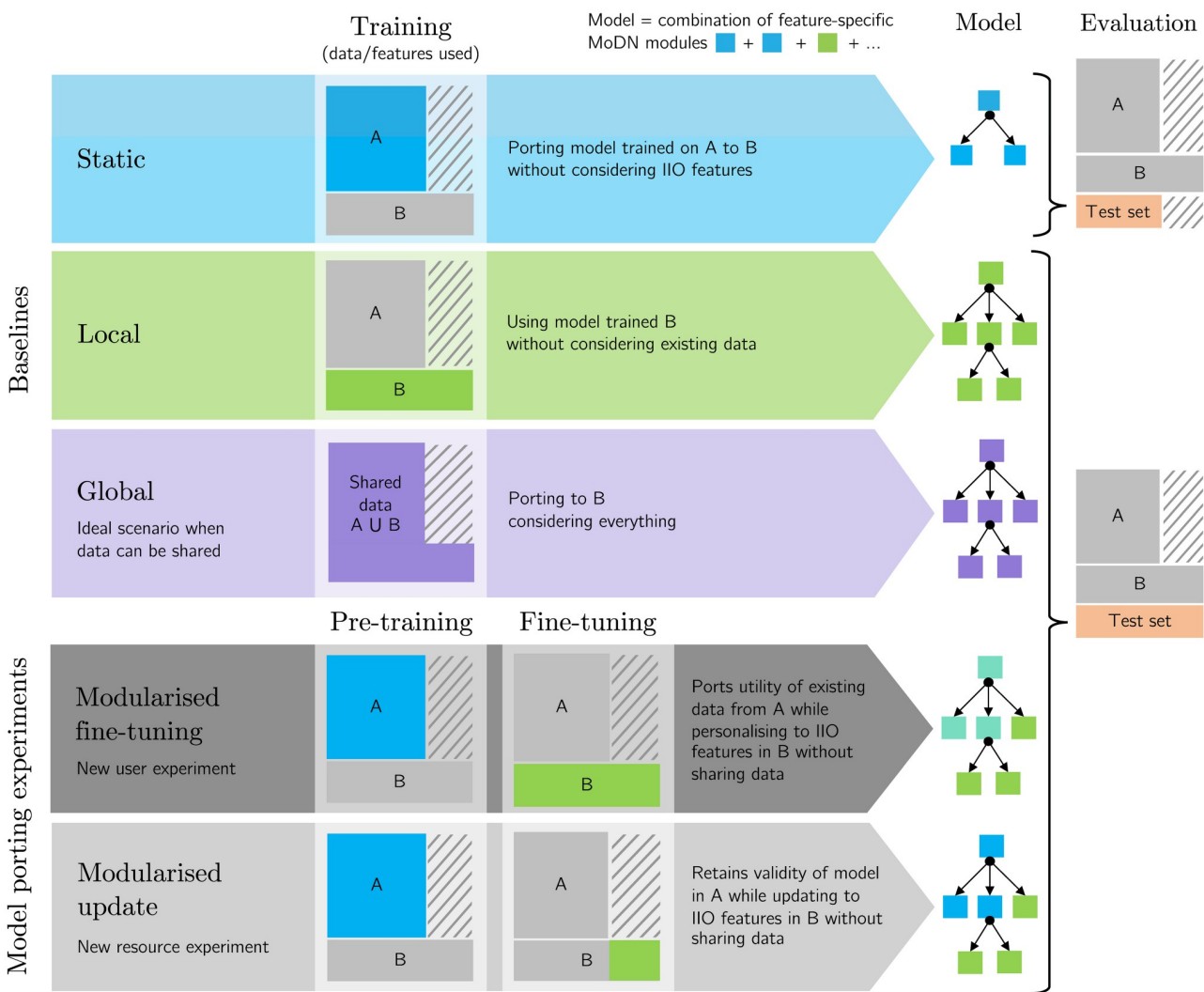

**Fig 2. Experimental set up for porting MoDN modules in IIO settings.** MoDN is tested in two "model porting experiments" (grey), where modules are ported from a larger *source* data set (**A**) for fine-tuning or updating on a smaller, imperfectly interoperable *target* data set (**B**). The two experiments represent either a scenario where a new user with different resources starts using a CDSS or where an existing user gains new resources and would like to merge training. Three baselines are proposed. **Static** (blue) where modules trained in **A** are directly tested in **B**, thus not considering additional IIO features. **Local** (green) where modules are only trained on the target data set **B**, thus without insights from the larger source data set. **Global** (purple) is the *ideal* but unlikely, scenario of when all data can be shared between **A** and **B** and the modules are trained on the union of data (**A** ∪ **B**). The **modularised fine-tuning** experiment, pre-trains on **A** and then fine-tunes all modules (for all features) on **B** (thus personalising the modules trained on **A**). The **modularised update** experiment, pre-trains the blue modules on **A** and then adds modules specific to the new IIO features (in green) which have been independently trained on **B** (thus preserving the validity of the modules trained on **A**). The colors of the MoDN modules illustrate their training on distinct data sets and their potential re-combination in the porting experiments. In particular, the modules trained on **A** (blue) and fine-tuned on **B** (green) are thus depicted in teal.

the prediction regardless of how many times it is asked) can be found in the supplement S1 Text.

## Visualizing the MoDN *state*

After encoding all available features and retrieving the resulting *states* from MoDN, we compute the t-SNE mapping of the 'vectorised patients' in order to visualise them as points on a

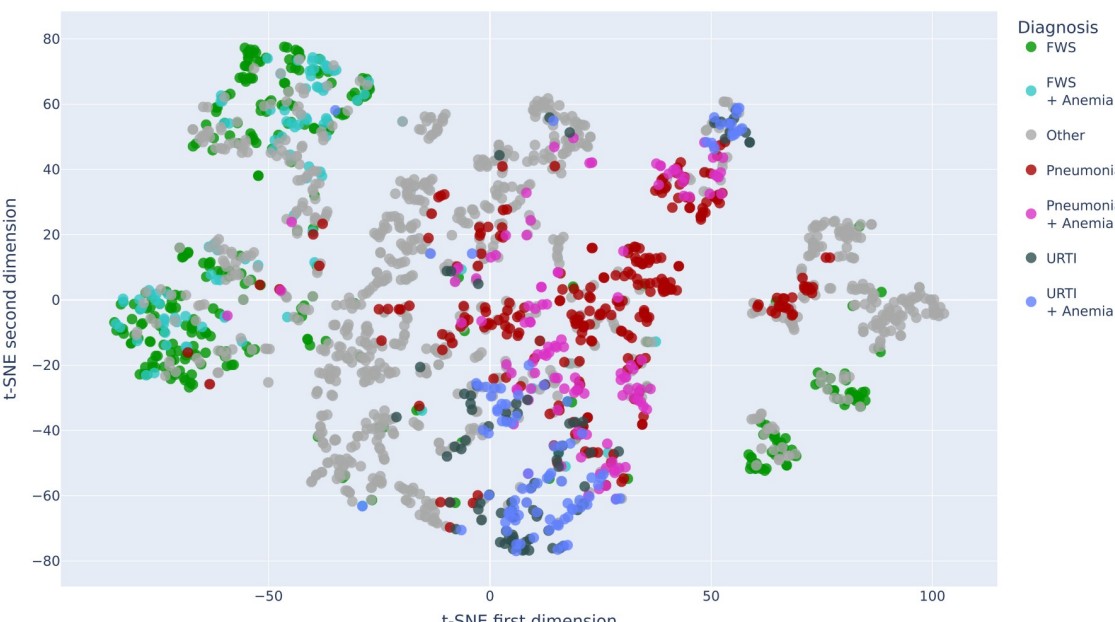

**Fig 3. Two dimensional t-SNE decomposition of the *state* vector for the patients of the training set.** The projection for each data point is overlaid with a color representing the true diagnosis/diagnoses of the patients. *URTI: Upper Respiratory Tract Infection, FWS: Fever Without Source.*

two-dimensional plot. For visual simplicity, we limit this to the patients in the training set who have one (or a combination) of the top 8 diagnoses, namely, pneumonia +/- anemia, fever without source (FWS) +/- anemia, upper respiratory tract infection (URTI) +/- anemia, diarrhea and 'other' (the latter of which is anticipated to have a more distributed placement on the plot).

Here, the *state* is represented as a point and clustered with similar *states*. Each mapped data point is colored according to its true diagnosis/diagnoses. Fig 3 shows several clearly homogenous clusters indicating that patients with the same diagnoses are 'close' to each other in our internal model representation (and thus that the *state* represents the outcome). Furthermore, we see smooth transitions between the clusters. For example, patients with FWS (in green) are mapped next to patients with a combination of FWS and anemia (in turquoise). Conversely, we see the *other* diagnoses to be more distributed in multiple clusters as expected.

## MoDN diagnosis decoding

The predictive performance of MoDN was compared to the best logistic regression and MLP algorithms for each target diagnosis. Fig 4 shows the macro $F_1$ scores (unweighted average of $F_1$ scores for the presence and absence of the disease). An overall performance is computed as the average over all diagnoses. Paired *t*-tests show that MoDN significantly outperforms the baselines for all binary classifications as well as for the overall diagnosis prediction. Malaria is an exception, where MoDN and baseline models have equivalent performance.

The confidence calibration plot in Fig 5 shows the predicted diagnosis probabilities by MoDN versus the correctness likelihood in the test set. For example, out of all the test points for which our model predicts a probability of having disease *d* of 0.9, our model is perfectly calibrated if 90% of these test points are indeed labeled as having *d*. In Fig 5, the points

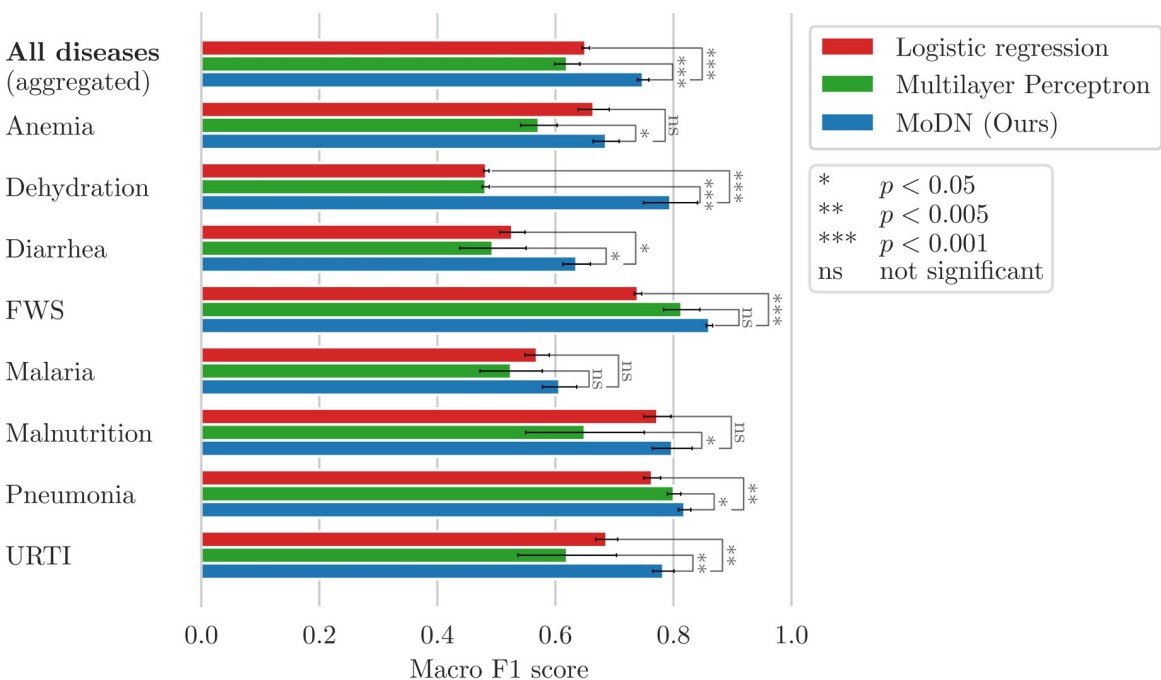

**Fig 4. MoDN diagnosis decoding performance.** Mean of the 5 × 2 cross-validated macro F1 scores for the diagnosis prediction on the test sets. Furthermore, MoDN significantly beats at least one of the baselines for each of the individual diagnoses except for malaria.

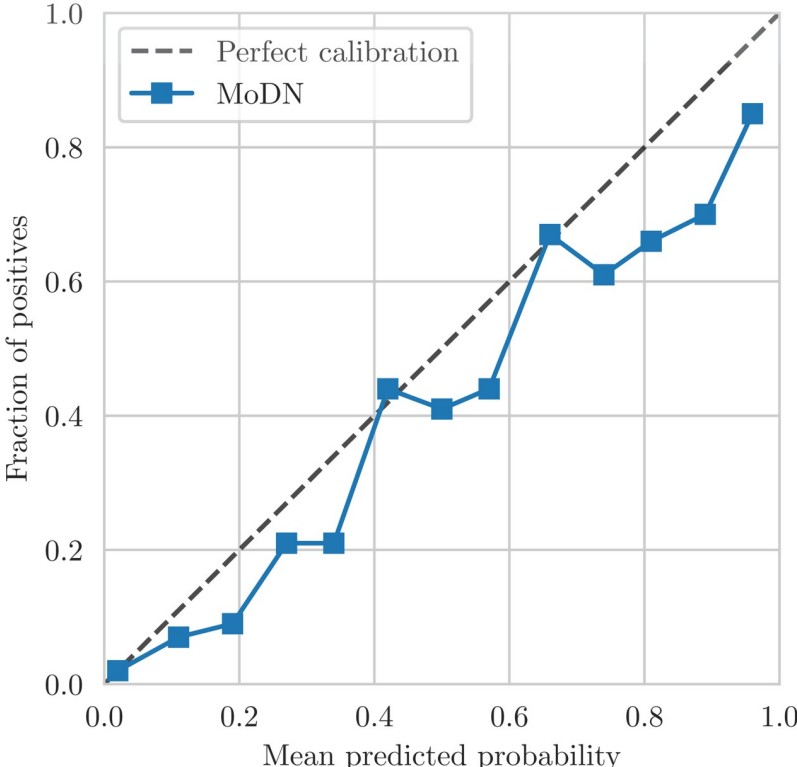

**Fig 5. MoDN calibration curve of the predictions on the test set after having encoded all available features.**

representing the confidence of our model are close to the line of perfect calibration. This shows that the predicted probabilities of MoDN are a good reflection of its confidence.

## Visualizing diagnostic trajectories

The metrics in the previous section show the model performance on unseen data once all the available features have been encoded Fig 4. One of the assets of our model is that it provides the clinician with feedback at any point in the consultation. For a given patient, the clinician can thus see how the predictions evolve as the features get encoded. The two heatmaps in Fig 6 show the predictive evolution for two randomly selected patients from the test set. The possible diagnoses are given by the $y$–axis and the $x$–axis shows the sequentially encoded features (from left to right), along with the values for that specific patient. We see how the model shifts towards the colour poles (blue and red extremes) as it learns more about the patient and gains confidence. The cut-off for binary classification is 50%. Thus, each percentage above 50% is increasing confidence in a positive diagnosis, while each percentage below is increasing confidence in a negative diagnosis. In each case, the model predicts the outcome correctly, but with various levels of confidence, with different patterns of evolution. For example, in Fig 6a predictive confidence accumulates slowly throughout the consultation, while in Fig 6b a confident prediction is achieved early, after a highly determinant question of "fever only". In another case (S3 Fig), the model predicts correctly but with less confidence (remaining at about 0.6). These feature-wise predictions thus give the clinician an assessment of the impact of that feature on the prediction.

## The new IIO user scenario: Modularised fine-tuning

Here, we test modularised fine-tuning of MoDN in the 'new IIO user' scenario as described in Fig 2. With MoDN, we can port the relevant modules pre-trained on the larger more established *source* data set to another smaller and IIO *new* data set and fine tune them whilst adding additional modules. Fig 7 shows the performance in macro $F_1$ score for this proposed solution (dark grey, *Modular fine-tuning*) tested in four levels of feature overlap (from 60—100%), and compared to three baselines described in Fig 2; i.e., global (training on shared data), local (training on full feature set but only on new data), and static (training only on source data and thus with a restricted feature set).

We see that a MoDN model built with modularised fine-tuning (without sharing data) matches the performance of the global model (trained on the union of shared data) and that it maintains its performance at all levels of feature overlap tested. This is in contrast to the static model (teal), which is significantly affected by decreasing feature overlap.

**The new IIO resource scenario: Modularised updating.**   This model isolates training updates to newly added features (in the scenario where new resources become available and are added to the decision tree). We use the trained modules from the source data set as a starting point and keep their parameters fixed. We then apply the frozen modules to the local data set and only train the modules corresponding to new features. The performance of this model is shown in light grey in Fig 7. Similarly to the modularised fine-tuning, we see that modularised update matches the global model where all data is shared. This shows that MoDN decision rules can be adapted to new features without modifying previously validated predictions of existing features, thus preserving the validity of the tool.

## Discussion

With the increasing use and complexity of electronic health records, there is enormous potential for deep learning to improve and personalise predictive medicine. However, it has not yet

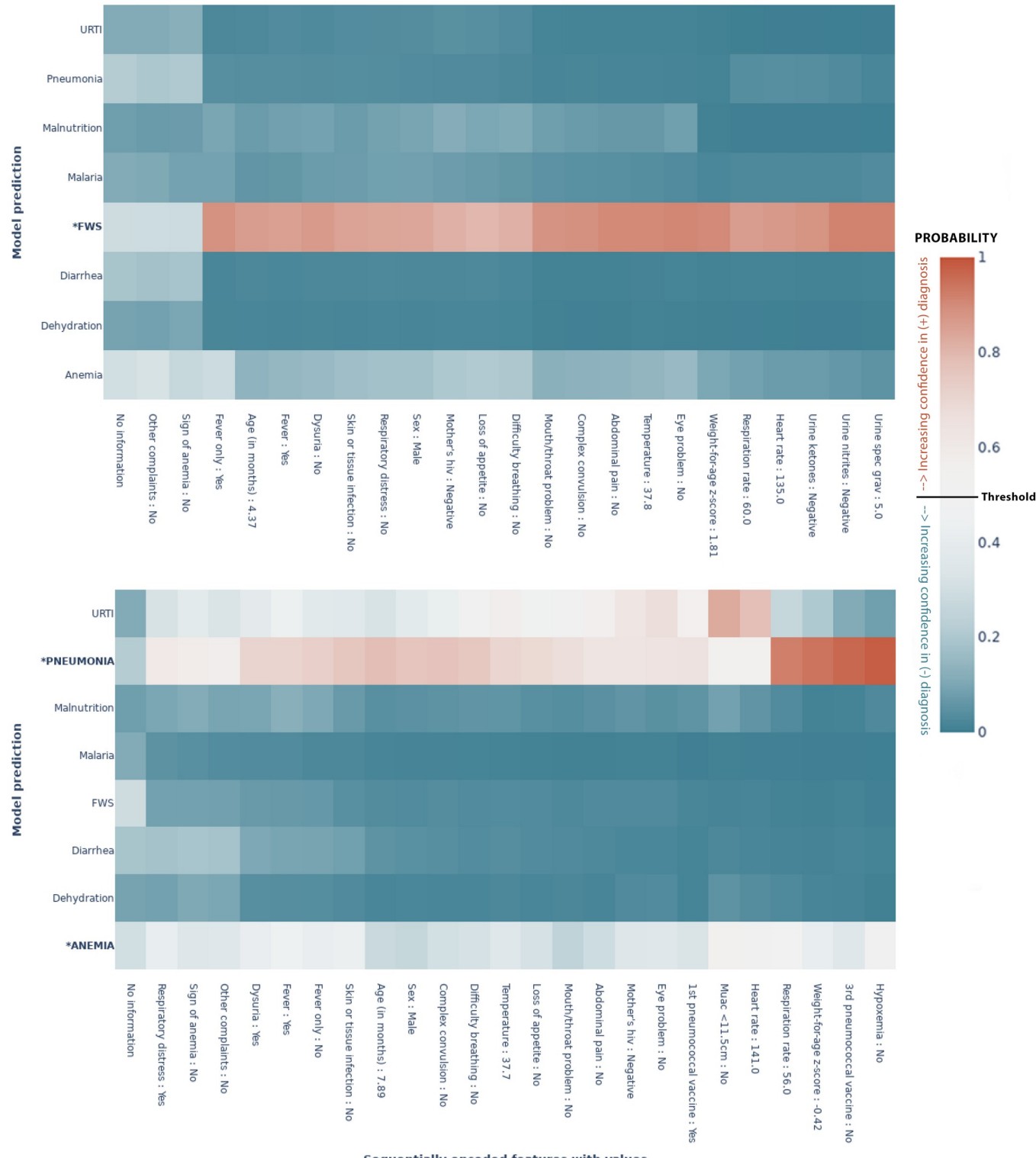

**Fig 6. MoDN's feature-wise predictive evolution in two random patients.** Each graph represents a single patient randomly selected from the test set. The $y$–axis lists the eight possible diagnoses predicted by our model. The true diagnosis of the patient is in bold and marked by an '*'. The $x$–axis is a sequential list of questions asked during the consultation (the response of that specific patient is also listed). In each case the model predicts the true label correctly. The heatmap represents a scale of predictive certainty from red (positive, has diagnosis) to blue (negative, does not have diagnosis), where white is uncertain. **(a)** Patient with the true diagnosis of pneumonia and anaemia. Here, predictive confidence accumulates slowly throughout the consultation. **(b)** Patient with a true diagnosis of

FWS. Here, a confident prediction is achieved early after a highly determinant question of "fever only". *: *True diagnosis, URTI: Upper Respiratory Tract Infection, FWS: Fever Without Source, Threshold: probability at which the model categorises the patient with a diagnosis(50%).*

reached wide-spread use due to fundamental limitations such as insufficient performance, poor interpretability, and the difficulty of validating a continuously evolving algorithm in prospective clinical trials [33, 34]. When deployed, there is a tendency to favor sparsely-featured linear models, probably for their inherent interpretability and as a consequence of exploiting fortuitous feature overlaps between imperfectly interoperable (IIO) data sets, which limits feature diversity. However, the personalising patterns in health data are unlikely to be linear nor explained by a few features [35].

In this work, we proposed MoDN, a novel approach to constructing decision support systems that seeks to address the issues above, allowing interpretable deep learning on IIO data sets.

In predicting diagnoses Fig 4 and features (supplement S1 Text), we see that the modularity of MoDN yields a significant performance benefit compared to its 'monolithic' counterparts, the latter of which process all features at once as opposed to an ensemble of feature-wise models. It outperforms logistic regression as well as MLP for all diagnoses excepting malaria. As malaria is strongly predicted by a single feature (i.e the rapid diagnostic test), it is anticipated that model design would have limited predictive value. Particularly useful is that the gain in

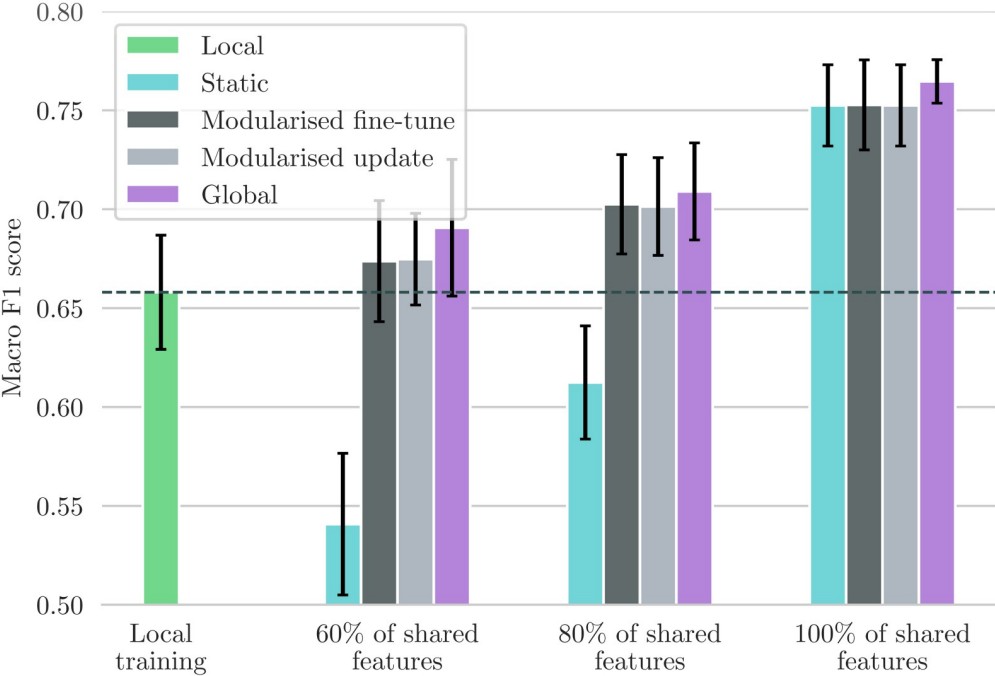

**Fig 7. Comparison between the ported models and the baselines.** Performance metric is the mean macro F1 scores with 95% CIs. Modularised fine-tuning or updating on additional local features (**gray**) consistently increases the model's performance compared to statically using a source model that only uses shared features (**teal**). The modularised update scenario achieves this without changing the model's behaviour on patients in the source dataset. The fine-tuning approaches perform almost as well as the global baseline (**purple**) that trains on the union of shared data. When the percentage of shared features is 80 or 100%, fine-tuning is significantly better than training only locally on the small 'target' dataset (**green**).

performance does not come at a higher computational cost, as MoDN uses a similar number of parameters.

The architecture of our model is similar to classic modular neural networks (MNNs) described by Shukla et al. [36]. However, there is little literature on the implementation of MNNs in the medical field and in the examples found, the module resolution is limited to pre-defined feature clusters such as in Pulido et al. [37] for the diagnosis of hypertension. The feature-wise modules in our proposed method means that no subgroups of features are constrained to be present together.

Visualising the *states* of MoDN in a 2-dimensional space Fig 3, we see the clustering and granularity of patient representations align with interpretable expectations of patient similarity. For instance, the catchall diagnoses of 'other' and 'fever without source' (FWS) have the most distributed spread, while more specific diagnoses have more homogeneous clusters. Interestingly for FWS, we see the states are distributed into four clear sub-populations, which we hypothesise may relate to the unknown etiology of the patient's fever. A previous effort to cluster patient profiles using unsupervised auto-encoders showed the potential of deriving a general-purpose patient representation from medical data and how it could facilitate clinical predictive modeling [38].

MoDN sequentially ensembles feature-specific modules into a continuously 'extending' model. A key benefit of this approach is the ability to make granular interrogations of the predictive impact of each feature. Thus, MoDN aligns the learning process with the clinician, where they can visualise the diagnosis evolve during the consultation. This could act as a training tool, helping them to understand the impact of their responses, which may in turn guide more careful collection of highly determinant features. To the best of our knowledge, the evolutive feature importance of MoDN is unique to the literature on CDSS. In traditional monolithic models, feature importance is computed retrospectively, using computationally expensive techniques, which may not allow the user to make corrective steps at the time of feature collection.

As stressed in many works, systematic missingness poses a major limitation to traditional deep learning on CDSS data [25]. Typical ML algorithms that operate on a vector of features are particularly affected by missingness because all data points must be encoded in some way in order to use the feature. The result is that the feature is either dropped or imputed, thus either reducing available information or injecting noise/bias. The risk of bias is particularly high when imputing systematically missing data common to CDSS. Thus, traditional models carry the risk of exploiting clinically irrelevant patterns of systematic missingness. The feature-specific modules of MoDN on the other hand, by design, cannot detect cross-feature patterns in missingness, and no imputation or feature limitations are required. Few other options exist for imputation-free neural nets. For instance, *Network Reduction* proposes a single neural net for each possible configuration of complete features [39]; an idea that has since been iterated by Krause et al. [40] and Baron et al. [41]. However, all these approaches suffer at scale, where the number of possible configurations grows exponentially with the number of features, creating an unfeasible computational overhead in high-dimensionality data sets. By comparison, the number of neural nets in our approach is linear in the number of features.

As mentioned throughout this work, MoDN seeks to use modularisation to address the issues faced during collaborative algorithm updates, i.e. 1) offering users the ability to update targeted modules in vacuo, thus retaining the validity of previously validated modules, and 2) allowing models to be ported between IIO datasets. A large-scale study on 200,000 patients by Google research [19], demonstrated an approach to deploying deep learning on IIO data from multiple centers. They developed an automated data harmonisation pipeline, and showed that they could accurately predict multiple medical events. However, such models would still be

limited to the fortuitously overlapping feature sets. The results in Fig 7 show that MoDN is able to address the loss of information caused by IIO data in decentralised settings, where it matches performance of a 'global' baseline trained on the union of shared data. This portability, also makes MoDN more amenable to distributed learning.

## Limitations

This work sought to validate MoDN on a real-world data set and specifically work within the consultation logic of the CDSS. This also limits its findings to the inherent diversity available in the question structure of this data set (albeit large, with over 200 unique question sets detected). A knock on effect of training MoDN on a single fixed questionnaire logic, is that we cannot guarantee the performance of the algorithm if the question order used in a consultation differs from the order in the training set.

This could be addressed by simply randomising all questions into a global question block, which would also allow MoDN to provide insights on the 'next most predictive questions to ask'. For the IIO experiments, we purposely use an experimental setup that mimics two imperfectly interoperable data sets (i.e. splitting data sets and random feature deletion). We use this instead of an independent IIO dataset to better isolate the effect of the IIO without influence from data-dependent variation in the distribution of each feature across two data sets. "External validation on a larger data set" is desirable in any study. However, we could not find any public patient-level CDSS-derived datasets on a comparable population, even with only a partial overlap of collected features.

## Conclusion

This work showcases the various advantages of modularising neural nets into input-specific modules. First *composability*—where the user can input any number or combination of questions and output any number or combination of predictions. Second, the *flexible portability* of the modules also provides more granular options to building collaborative models which may address some of the most common issues of data ownership and privacy. Third, modularity also creates an *inherent interpretabilty* where the sequential deployment of modules provides granular, input-specific interpretable feedback that aligns with the sequential logic of a medical consultation. Finally, MoDN's ability to *skip over missing inputs* enables it to train on CDSS-derived data irrespective of the presence of biased missingness, and thus for it to be used flexibly across settings with evolving access to resources.

## Supporting information

**S1 Table. Summary statistics of the features of the e-POCT data set.**
(PDF)

**S1 Text. Implementation details, feature decoding, idempotence and calibration.**
(PDF)

**S1 Fig. Macro F1 scores for the disease prediction on test set, when the model is additionally trained to perform feature decoding.** The baselines of MLP and logistic regression with $L_2$ penalty were tuned to achieve maximal performance. MoDN outperforms the baselines significantly for the overall disease prediction. Furthermore, it outperforms the performance of at least one of the baselines for each of the individual diseases, except for pneumonia.
(TIFF)

**S2 Fig. MoDN calibration curve of the predictions on the test set after having encoded all available features, when the model is additionally trained for feature decoding.** MoDN is calibrated close to the perfect calibration line.
(TIFF)

**S3 Fig. MoDN's feature-wise predictive evolution in a random patient.** This graph represents a single patient randomly selected from the test set. The $y$–axis lists the eight possible diagnoses predicted by our model. The true diagnosis of the patient is in bold and marked by an '*'. The $x$–axis is a sequential list of questions asked during the consultation (the response of that specific patient is also listed). In each case the model predicts the true label correctly. The heatmap represents a scale of predictive certainty from red (positive, has diagnosis) to blue (negative, does not have diagnosis), where white is uncertain. This patient has a true diagnosis of FWS and anemia. The model predicts these correctly but with less confidence, as can be interpreted from lighter colours. *: *True diagnosis, URTI: Upper Respiratory Tract Infection, FWS: Fever Without Source.*
(TIFF)

**S4 Fig. AUC-ROC curves for diagnosis.** Mean and standard deviation of the AUC and ROC curve for diagnosis prediction computed on the test set, using bootstrapping.
(TIFF)

## Acknowledgments

The authors thank the patients and caregivers who made the study possible, as well as the clinicians who collected the data on which MoDN was validated [29].

## Author Contributions

**Conceptualization:** Thijs Vogels, Mary-Anne Hartley.

**Data curation:** Cécile Trottet, Kristina Keitel, Mary-Anne Hartley.

**Formal analysis:** Cécile Trottet.

**Funding acquisition:** Mary-Anne Hartley.

**Investigation:** Cécile Trottet, Thijs Vogels, Mary-Anne Hartley.

**Methodology:** Cécile Trottet, Thijs Vogels, Mary-Anne Hartley.

**Project administration:** Martin Jaggi, Mary-Anne Hartley.

**Resources:** Kristina Keitel, Martin Jaggi.

**Supervision:** Thijs Vogels, Kristina Keitel, Alexandra V. Kulinkina, Rainer Tan, Ludovico Cobuccio, Martin Jaggi, Mary-Anne Hartley.

**Validation:** Cécile Trottet, Thijs Vogels.

**Visualization:** Cécile Trottet, Thijs Vogels, Mary-Anne Hartley.

**Writing – original draft:** Cécile Trottet, Thijs Vogels, Mary-Anne Hartley.

**Writing – review & editing:** Cécile Trottet, Thijs Vogels, Alexandra V. Kulinkina, Rainer Tan, Ludovico Cobuccio, Martin Jaggi, Mary-Anne Hartley.

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
