## [Decision Letter · Decision Letter 0]

26 Jan 2023

PDIG-D-22-00242

Modular Clinical Decision Support Networks (MoDN)—Updatable, Interpretable, and Portable Predictions for Evolving Clinical Environments

PLOS Digital Health

Dear Dr. Hartley,

Thank you for submitting your manuscript to PLOS Digital Health. After careful consideration, we feel that it has merit but does not fully meet PLOS Digital Health's publication criteria as it currently stands. Therefore, we invite you to submit a revised version of the manuscript that addresses the points raised during the review process.

Please submit your revised manuscript within 60 days Mar 27 2023 11:59PM. If you will need more time than this to complete your revisions, please reply to this message or contact the journal office at digitalhealth@plos.org. Please include the following items when submitting your revised manuscript:

We look forward to receiving your revised manuscript.

Kind regards,

Henry Horng-Shing Lu

Section Editor

PLOS Digital Health

Journal Requirements:

b. If any authors received a salary from any of your funders, please state which authors and which funders.

2. We ask that a manuscript source file is provided at Revision. Please upload your manuscript file as a .doc, .docx, .rtf or .tex.

3. Please provide separate figure files in .tif or .eps format only and remove any figures embedded in your manuscript file. Please also ensure that all files are under our size limit of 10MB.

4. We have noticed that you have uploaded Supporting Information files, but you have not included a list of legends. Please add a full list of legends for your Supporting Information files after the references list.

Additional Editor Comments (if provided):

Reviewers' comments:

Reviewer's Responses to Questions

**Comments to the Author**

1. Does this manuscript meet PLOS Digital Health’s publication criteria? Is the manuscript technically sound, and do the data support the conclusions? The manuscript must describe methodologically and ethically rigorous research with conclusions that are appropriately drawn based on the data presented.

Reviewer #1: Yes

Reviewer #2: Yes

2. Has the statistical analysis been performed appropriately and rigorously?

Reviewer #1: Yes

Reviewer #2: I don't know

3. Have the authors made all data underlying the findings in their manuscript fully available (please refer to the Data Availability Statement at the start of the manuscript PDF file)?

Reviewer #1: Yes

Reviewer #2: No

4. Is the manuscript presented in an intelligible fashion and written in standard English?

Reviewer #1: Yes

Reviewer #2: Yes

5. Review Comments to the Author

Reviewer #1: The authors present a development/validation study of a novel CDSS for diagnosis of eight conditions among pediatric outpatients in Tanzania. The CDSS, MoDN, outperforms the authors' chosen baseline models and appears to be relatively robust to "new" (simulated) environments where interoperability may be an issue. The authors explain their modelling technique well, and the use of good figures and illustrative examples benefit the manuscript greatly. 

Minor comments:

- In my opinion, the baseline models seem to be set up for failure from the start. For example, the authors discuss the benefits of informative missingness, then choose to perform mean-imputation to generate the training data for the baseline models. Why not encode missingness? Why not use a type of model that generally excels with tabular data and multiclass prediction such as gradient-boosting trees/random forest?

- Some of the hyperlinks are confusingly labelled, e.g. line 221 refers to "1 and 1" where each "1" hyperlinks to a different section of the manuscript.

- Lines 250-252/Figure 5: the authors state the points are "close to the line of perfect calibration" - what does "close" mean? It looks quite good, it goes in the right direction, but of the 13 points, 10 are below the diagonal which would suggest the model systematically over-predicts. It would be better to quantify this using e.g. the Brier-score loss or similar. The way binning is performed will also have some impact on how the calibration curve looks, especially for smaller datasets.

- Lines 336-345: sequential/continuous updating of beliefs is also a feature of Bayesian Networks: prior beliefs are updated based on conditional probabilistic relationships (e.g. learned model parameters) and the evidence input and allow the impact of each piece of evidence to be quantified. Feature importance for traditional statistical models such as LR is also easily quantified in real time.

- Tab S1: adjusting the precision of the min and max values for each feature would make the table much more readable. The description of "complaint" is NaN. Various formatting issues with use of D0, DO and d0.

Reviewer #2: This is an interesting manuscript presenting an approach how MoDN can achieve to give feedback to the clinician for Clinical Decision Support. As an researcher/clinical pharmacologist I will not comment on the methodological aspects yet focus my comments on the practical aspects including useability of CDSS and meaning of these findings for the clinician. 

My main comment is that this paper does not – as the title suggests – make clear what this solution means with respect to updating knowledge and interpretation in clinical care. Hence, it is not clear how this work will help the clinician solving actual problems. My suggestion would be to add this information to this manuscript (methods, results and discussion). My detailed comments are:

1. The aim of this study is to present interpretable and predictive feedback to the clinicial. First, please explain how your CDSS works in the routine of the clinicial. At what moment will she/he consult CDSS? Are extra data needed?

2. You aim to support the clinician to predict the development of 8 diagnoses: anaemia, dehydration, malaria, diarrhoea, fever, malnutrition, pneumonia, upper respiratory tract infection. Can you please explain how your system is meant to work. 

a. Please elobarate on how you support clinicians. How far before the actual diagnose are they supported. Can your system predict events that will occur in two months (I doubt). Or can they predict malaria once there is a diagnostic test available (in that situation: what is the meaning of CDSS as you already have a diagnosis). It is not clear for me how your system is supposed to work.

b. What is the predictive value of your system? Sensitivity and specificity. As a clinician I want to be really sure that I will not miss a malaria/pleumonia.

c. What are the requirements with respect to patient data needed. Are there any restrictions with respect to the (patient) data needed? E.g. is the system also accurate as there is no diagnostic test for malaria? Or haemoglobin for anemia?

3. Introduction. Some sentences might be somewhat speculative.

a. In the introduction (line 11-16) authors make some very general assumptions about CDSS in a way that seems to disqualify lack of available evidence, and use this as motivation for their approach. From a clinical perspective, also expert opinion – for example by elaboration of clinical or pharmacological knowledge (Weersink, BMJ Open; Van Tongeren, Frontiers) – is an important approach to develop the best possible recommendations. The context should further motivate the best approach. In most situations clinicians favor an approach where they can understand why certain options in clinical decision making are preferred in comparison to others.

4. Methodology. I am not an expert in this field, but please state that the dataset that is used is big enough for the analysis you used and the data contain enough details to analyse this, and that the outcomes can/cannot be used in other settings. Do you aim to develop general CDS rules for all pediatrics in the world, or only for the local setting?

5. Should the model not be tested on a datasource outside the 3192 outpatients in another setting?

Discussion

6. Can you discuss in what situation this model will help pediatricians in their clinical care? What tasks are supported? To what extend and for what diagnoses will this help the patient?

6. PLOS authors have the option to publish the peer review history of their article (what does this mean?). If published, this will include your full peer review and any attached files.

**Do you want your identity to be public for this peer review?** For information about this choice, including consent withdrawal, please see our Privacy Policy.

Reviewer #1: No

Reviewer #2: No

---

## [Decision Letter · Decision Letter 1]

25 Apr 2023

PDIG-D-22-00242R1

Modular Clinical Decision Support Networks (MoDN)—Updatable, Interpretable, and Portable Predictions for Evolving Clinical Environments

PLOS Digital Health

Dear Dr. Hartley,

Thank you for submitting your manuscript to PLOS Digital Health. After careful consideration, we feel that it has merit but does not fully meet PLOS Digital Health's publication criteria as it currently stands. Therefore, we invite you to submit a revised version of the manuscript that addresses the points raised during the review process.

Please submit your revised manuscript within 30 days May 25 2023 11:59PM. If you will need more time than this to complete your revisions, please reply to this message or contact the journal office at digitalhealth@plos.org. Please include the following items when submitting your revised manuscript:

We look forward to receiving your revised manuscript.

Kind regards,

Henry Horng-Shing Lu

Section Editor

PLOS Digital Health

Journal Requirements:

Additional Editor Comments (if provided):

Reviewers' comments:

Reviewer's Responses to Questions

**Comments to the Author**

1. If the authors have adequately addressed your comments raised in a previous round of review and you feel that this manuscript is now acceptable for publication, you may indicate that here to bypass the “Comments to the Author” section, enter your conflict of interest statement in the “Confidential to Editor” section, and submit your "Accept" recommendation.

Reviewer #1: All comments have been addressed

Reviewer #2: (No Response)

2. Does this manuscript meet PLOS Digital Health’s publication criteria? Is the manuscript technically sound, and do the data support the conclusions? The manuscript must describe methodologically and ethically rigorous research with conclusions that are appropriately drawn based on the data presented.

Reviewer #1: Yes

Reviewer #2: Yes

3. Has the statistical analysis been performed appropriately and rigorously?

Reviewer #1: Yes

Reviewer #2: I don't know

4. Have the authors made all data underlying the findings in their manuscript fully available (please refer to the Data Availability Statement at the start of the manuscript PDF file)?

Reviewer #1: Yes

Reviewer #2: No

5. Is the manuscript presented in an intelligible fashion and written in standard English?

Reviewer #1: Yes

Reviewer #2: Yes

6. Review Comments to the Author

Reviewer #1: Thank you for the opportunity to review the revised manuscript. All my comments have been successfully addressed and I commend the authors on their work.

Reviewer #2: First, I would like to thank the authors for answering the questions and the improved version of the manuscript. An issue that has not been solved, and what would improve the clinical applicability of the manuscript is a description how this tool is to be used in the patient journey. Yet is is difficult to understand the added value for clinical practice. In this descriptions the authors should show how this tool is or should be implemented in clinical care, so that it is clear what other steps or actions can be skipped or improved. 

Some other questions I asked have been answered adequately, yet no changes are made in the manuscript. I believe that the manuscript will be improved if these answers will also be included in the manuscript (e.g. questions 2a - how you support clinicians; 2c - requirements with respect to patient data needed; 5. external validation; include as limitation in the text; 6. explain in what situations this model will help.

Technical remarks: 

- I did not receive a track changes version that made it difficult to check the changes in detail.

- the link provided by the authors with the underlying data could not be opened by me. I could see there was a file posted, yet I was unable to check if this file included the underlying data. I believe the editorial office will further check check this.

7. PLOS authors have the option to publish the peer review history of their article (what does this mean?). If published, this will include your full peer review and any attached files.

**Do you want your identity to be public for this peer review?** For information about this choice, including consent withdrawal, please see our Privacy Policy. 

Reviewer #1: No

Reviewer #2: Yes: Sander Borgsteede

---

## [Decision Letter · Decision Letter 2]

12 Jun 2023

Modular Clinical Decision Support Networks (MoDN)—Updatable, Interpretable, and Portable Predictions for Evolving Clinical Environments

PDIG-D-22-00242R2

Dear Dr Hartley,

We are pleased to inform you that your manuscript 'Modular Clinical Decision Support Networks (MoDN)—Updatable, Interpretable, and Portable Predictions for Evolving Clinical Environments' has been provisionally accepted for publication in PLOS Digital Health.

Best regards,

Henry Horng-Shing Lu

Section Editor

PLOS Digital Health

Reviewer Comments (if any, and for reference):

Reviewer's Responses to Questions

**Comments to the Author**

1. If the authors have adequately addressed your comments raised in a previous round of review and you feel that this manuscript is now acceptable for publication, you may indicate that here to bypass the “Comments to the Author” section, enter your conflict of interest statement in the “Confidential to Editor” section, and submit your "Accept" recommendation.

Reviewer #1: All comments have been addressed

Reviewer #2: All comments have been addressed

2. Does this manuscript meet PLOS Digital Health’s publication criteria? Is the manuscript technically sound, and do the data support the conclusions? The manuscript must describe methodologically and ethically rigorous research with conclusions that are appropriately drawn based on the data presented.

Reviewer #1: Yes

Reviewer #2: Yes

3. Has the statistical analysis been performed appropriately and rigorously?

Reviewer #1: Yes

Reviewer #2: I don't know

4. Have the authors made all data underlying the findings in their manuscript fully available (please refer to the Data Availability Statement at the start of the manuscript PDF file)?

Reviewer #1: Yes

Reviewer #2: Yes

5. Is the manuscript presented in an intelligible fashion and written in standard English?

Reviewer #1: No

Reviewer #2: Yes

6. Review Comments to the Author

Reviewer #1: Thank you again for the chance to review this manuscript. I have no further general comments about the text, from what I can see all comments have been addressed adequately. There was a minor error in the github link to the source code, in the manuscript it appears as https://github.com/epfliglobalhealth/MoDN-TrottetVogels2022 as opposed to the correct

https://github.com/epfl-iglobalhealth/PLOSDH-MoDN-TrottetVogels2022

Reviewer #2: I would like to thank the authors for their dedication to this subject. I am still not happy with the clinical perspective. The paper would improve if you could describe from the perspective of doctor or patients when he encounters your tool. At first consultation, the doctor will use this tool? And decisions will be made based on this tool? During hospital stay nurses walk around to use this tool for every new admission? The practical implication still is vague

7. PLOS authors have the option to publish the peer review history of their article (what does this mean?). If published, this will include your full peer review and any attached files.

**Do you want your identity to be public for this peer review?** For information about this choice, including consent withdrawal, please see our Privacy Policy.

Reviewer #1: No

Reviewer #2: **Yes: **Borgsteede, S.D.
